# Study on Electron-Induced Surface Plasmon Coupling with Quantum Well Using a Perturbation Method

**DOI:** 10.3390/nano10050913

**Published:** 2020-05-09

**Authors:** Yifan Chen, Yulong Feng, Zhizhong Chen, Fei Jiao, Jinglin Zhan, Yiyong Chen, Jingxin Nie, Zuojian Pan, Xiangning Kang, Shunfeng Li, Qi Wang, Shulin Zhang, Guoyi Zhang, Bo Shen

**Affiliations:** 1State Key Laboratory for Artificial Microstructure and Mesoscopic Physics, School of Physics, Peking University, Beijing 100871, China; 1501110129@pku.edu.cn (Y.C.); fengyulong@pku.edu.cn (Y.F.); fjiao@pku.edu.cn (F.J.); 1601110180@pku.edu.cn (J.Z.); chenyiyong@pku.edu.cn (Y.C.); niejingxin@pku.edu.cn (J.N.); ipanzj@whu.edu.cn (Z.P.); xnkang@pku.edu.cn (X.K.); slzhang@pku.edu.cn (S.Z.); gyzhang@pku.edu.cn (G.Z.); bshen@pku.edu.cn (B.S.); 2Institute for Quantum Science and Engineering, Southern University of Science and Technology, Shenzhen 518055, China; 3State Key Laboratory of Nuclear Physics and Technology, School of Physics, Peking University, Beijing 100871, China; 4Dongguan Institute of Optoelectronics, Peking University, Dongguan 523808, China; lisf@pku-ioe.cn (S.L.); wangq@pku-ioe.cn (Q.W.)

**Keywords:** localized surface plasmon, green LED, cathodoluminescence, FDTD, perturbation method

## Abstract

Ag nanoparticles (NPs) are filled in a photonic crystal (PhC) hole array on green light emitting diodes (LEDs). The localized surface plasmon (LSP)–quantum well (QW) coupling effect is studied by measuring the cathodoluminescence (CL) spectra impinging at the specific spots on the Ag NPs. Twenty-six percent and fifty-two percent enhancements of the CL intensities are obtained at the center and edge of the Ag NP, respectively, compared to the result that the electron-beam (e-beam) excites the QW directly. To illustrate the coupling process of the three-body system of e-beam–LSP–QW, a perturbation theory combining a three-dimensional (3D) finite difference time domain (FDTD) simulation is put forward. The effects of the polarization orientation of the dipole and the field symmetry of the LSP on the LSP–QW coupling are also discussed.

## 1. Introduction

Recently, surface plasmon (SP) shows highly potential applications in high efficiency and high speed light emitting devices for its coupling to the excitons in radiators and/or the photons in free space [1,2,3]. When the SP is resonantly excited in the metallic nanostructures, its plasmonic cavity is formed, in which the spontaneous emission rate (SER) of the radiators can be enhanced more than 1000 times [4,5]. After SP is excited in the metallic geometries of bow tie, nanopatch, nanoarch, and nanoparticle arrays, the plasmonic antennas efficiently extracted the near-field energy into the free space, coherently or in directional [5,6,7,8,9,10]. Low efficiency devices emitting green, ultraviolet light, or with efficiency droop under high injection level will be remedied by SP techniques [11,12,13,14]. The color conversion enhancement for quantum dots or fluorescent molecules on micro light emitting diodes (LEDs) are promptly required for SP coupling by resonant absorption and emission enhancement [15,16]. The modulation bandwidth of LEDs will be significantly increased to tens of GHz with the SER enhancement [17,18,19]. The smaller power dissipation and faster speed than lasers can reduce the energy budget of light interconnection to about several fJ/bit [2]. However, the photoluminescence (PL) intensity was suppressed two orders even though the SER enhanced 55 times for coupling of SP with InGaN quantum wells (QWs) [4]. If the energy dissipation in the metal is larger than the increase of the radiative recombination energy in radiators, the external quantum efficiency (EQE) will be reduced when the SP coupling happens. Many researchers have noticed the importance of combining the resonator and antenna by designing metal geometries to scatter the near-field energy out of the coupling system [6,10,20,21].

Energy transferring in the plasmonic system has been paid much attention to for revealing the coupling mechanisms between SP and radiators [1,13,22,23,24,25]. The total input energy transfers to SP radiating energy, SP dissipation energy, and radiators’ radiative and non-radiative recombination energies. On the one hand, the common individual metallic nanoparticle (NP) acts as a good resonator, not as a good antenna, so the most of the energies are transferred to the high-order modes of SP and dissipated by Ohmic loss [10]. On the other hand, the efficient antennas only enhance the excitation of SPs and light extraction, not significantly for the SER of the radiator [20,21]. Many antennas are fundamental prototypes, which may be hard to be the applicable ones [5,6]. The best practical results are only two-times enhancement for electroluminescence (EL) intensities [26] and modulation bandwidth [17] using the SP coupling to the InGaN QWs. Finite element method (FEM) numerical simulations are widely used to study on the coupling of SP and radiators [13,15,25,27,28,29,30]. In the simulation model, the radiators are often approximated as simple dipoles. The effects of the size, shape, arrangement of the metallic NPs and oscillating direction, location, amplitude, multiple radiators of dipoles on the coupling of SP, and radiators are simulated and are able to explain the experimental results well [13,15,27,28,29,30]. However, the energy transferring is not very clear for the localized SP (LSP) coupling to the multiple-dipole system, where the radiators show obviously different [15], or the collective resonance of the dipoles is significant [23,28].

In our previous work, we have studied on the energy transferring in the SP coupling to the systems of parallel and orthogonal dipoles [13,29,30,31]. In the case of parallel dipoles, the collective resonant effects on the internal quantum efficiency (IQE) and light extraction efficiency (LEE) are obtained. When the coupling system consists of more than one radiators, the radiators can interact with each other through the SP induced by themselves. The coupling effect on each dipole can hardly be separated from the whole system. Therefore, the energy transferring in the individual dipole and its corresponding SP radiation and dissipation energies are not known [31]. In the case of orthogonal dipoles, we distinguished the energy transferring in the individual dipole and its corresponding SP radiation and dissipation energies using the linear approximation [13]. Because the oscillating amplitude of the z-dipole is much larger than that of the x-dipole, the approximation is made that the dissipation energy and scattered energy of the z-dipole linearly changes with the total energy of the z-dipole. Then the energy transferring of the three-body system is revealed, and the cathodluminescence (CL) enhancement is explained in a green LED embedded with Ag NPs [13]. However, in many situations the oscillating amplitudes of dipoles are similar to each other. The linear approximation may cause a large error, for example, to a quantum dot (QD)-metallic NP–QW system [15,16]. To study on the energy transferring in such a three-body system, a perturbation method is proposed by adding a small power value, δP and its one-order asymptotic expansion. Theoretically, the perturbation theory is a set of methods for studying various problems in mathematics, mechanics, physics, and so on [32]. Moreover, the perturbation theory has been successfully applied to celestial mechanics (to study the Moon–Earth–Sun system) [33] and quantum mechanics [34]. In this work, the three-body system still consisted of a high energy electron-beam (e-beam), Ag NP, and green QWs, as well as that in Reference [13]. The e-beam has been dealt to be a dipole source [35,36]. Because CL measurements combine an ultrahigh spatial resolution of an electron microscope with a broadband optical sensitivity, they can be used to study the optical process in metal NPs [37]. The LSP–QW coupling processes under the influence of the LSP induced by a continuously injected e-beam are studied by CL spot scans. To illustrate the e-beam-influenced LSP–QW coupling mechanism, a perturbation calculation is performed combining with a three-dimensional finite difference time domain (3D-FDTD) (Lumerical FDTD Solutions 2020a, Vancouver, BC, Canada) method [38].

## 2. Experimental

The GaN-based green LED with a peak wavelength of 551 nm used in the experiment was grown on a c-plane sapphire substrate by metal organic chemical vapor deposition (MOCVD). The LED structure consists of a 4 μm n-GaN layer, 10 pairs of InGaN/GaN (2.5 nm/17.5 nm) QWs and a 180 nm p-GaN layer along the growth direction. The Ag NPs embedded in the hexagonal photonic crystals (PhCs) array holes were fabricated in the green LED by nanoimprint and lift-off techniques (labeled as Ag–PhC/QW), as described in Reference [29] in detail. The depth, diameter, and spatial period of the PhC structure were 170 nm, 600 nm, and 1 μm, respectively. Figure 1a shows the scanning electron microscope (SEM) image for the Ag–PhC/QW sample. The SEM images were recorded using an FEI NanoSEM 430 (FEI, Hillsboro, OR, USA). The diameter of the Ag NPs are 200 ± 20 nm. As a comparison, the sample with the same pattern but without QWs were fabricated on a GaN template (labeled as Ag–PhC/woQW).

A Gatan MonoCL4 system (Gatan, Pleasanton, CA, USA) was equipped in a SEM platform for the CL measurement as shown in Figure 1b schematically. The e-beam was highly focused and directly penetrated into the surface of the sample and excited the CL signal. The emitted light was collected by a parabolic mirror and detected by a charge-coupled device (CCD). To make sure that the emitted light can be collected at the maximum efficiency, the sample was placed at the focal plane of the parabolic mirror. When the e-beam impinged on the Ag NP, the collective electron oscillations in the metallic particle, namely the LSP, were induced and responsible for the light emission excited in the QWs underneath.

## 3. Results and Discussion

Figure 2 shows the CL spectra for the Ag–PhC samples with and without QWs measured at three different points in the PhC holes, which are marked as A, B, and C in the insets of the SEM images. The e-beam with an acceleration voltage of 10 kV is focused on Points A, B and C, respectively. The interval of the points is about 90 nm. In Figure 2a, there is only one peak wavelength at about 551 nm in each curve, corresponding to the emissions from QWs and/or LSPs. The LSPs are excited by the e-beam (ebLSP) and QWs (qwLSP). The CL peak intensities are 1.26 and 1.52 times for Points A and B compared with that for Point C. In Figure 2b, there are two peaks located at 360 and 580 nm, which are the band-edge emission and yellow luminescence (YL) of GaN, respectively. It also includes the emissions from the ebLSP and GaN excited LSP. At Points B and C, the CL spectra almost overlap. As to Point A, the YL intensity is about 60% compared with those for Points B and C. Moreover, the band-edge emission of GaN is hardly measured at Point A. It is obvious that the CL intensities in the Ag–PhC/QW sample are nearly two orders higher than those in the Ag–PhC/woQW sample. At Point A in Figure 2a, the e-beam can hardly penetrate through the Ag NP with the diameter of 200 nm [30]. The strong CL emissions mostly come from the LSP excited QW radiators under the Ag NP because the direct emission from the LSP is very weak [39]. Furthermore, it can be evidenced by the CL emission at Point A in Figure 2b, which is two orders less than that from QWs in Figure 2a. It also indicates that the emission of GaN excited by LSP is weak too. When the ebLSP couples with the QWs at Points A and B, the CL intensities are higher than that directly excited by the e-beam at Point C. At Point C, both the e-beam and the QWs are out of the evanescent field of LSP, whose featured length is given by [40],
(1)$$L=\frac{\lambda }{2\pi }\cdot \sqrt{\frac{\left(\varepsilon_{GaN}^{\prime }-\varepsilon_{metal}^{\prime }\right)}{{\varepsilon_{metal}^{\prime }}^{2}}}$$
where $\varepsilon_{GaN}^{\prime }$ and $\varepsilon_{metal}^{\prime }$ are the real parts of the dielectric constants of the semiconductor and metal. For GaN and Ag, L can be calculated as about 50 nm, indicating that the LSP-e-beam or LSP–QW coupling is weak at Point C. Therefore, the CL intensity at Point C is less than that at Point A in Figure 2a. As to the CL spectra at Point B, the CL intensity is strongest in Figure 2a. The CL enhancement may be due to the more LEE at the edge of the Ag NP since the light from the QWs is less sheltered. It will be discussed in the following FDTD simulations. In Figure 2b, since the CL enhancement by LSP coupling with radiators is small, the LEE enhancement at Point B cannot surpass the energy loss in the metal. In short, the coupling strength of LSP with the radiators may correspond to the intrinsic efficiency of the radiator and coupling conditions. The YL and GaN band edge radiators show less coupling strength to LSP than the QWs do. Generally the emission from InGaN QWs is two orders stronger than that from GaN bulk material under the same injection level.

To get a deeper understanding of the three-body (the e-beam–LSP–QW) coupling process, a 3D FDTD numerical method is used [38], where Maxwell’s equations are solved in a discretized space and time. Figure 3 shows the schematic simulation model. The QW is represented by a point dipole (labeled as q-dipole) polarized within the QW-plane (x–y plane) [13,25,27,29]. The polarized angle between the q-dipole orientation and x-axis, φ is set to vary from 0° to 180°. The spacing between the q-dipole and the bottom surface of the Ag NP is set to be 20 nm, which is consistent with the experimental condition. In FDTD simulations, the e-beam is usually modeled as a series of point dipoles with phase delay related to the e-beam velocity [35,39,41,42]. Since the e-beam velocity only modifies the phase delay through a cosine function, this allows one to split these dipoles into smaller sub-simulations. Cao, Y. et al. noted that the electron energy-loss spectroscopy, mainly determined by the electric field, of an e-beam interacting with metallic nanostructures can be related to the diagonal component of a Green tensor of Maxwell’s equations in the z direction [36]. The component of the Green tensor can be approximately obtained by placing an electric dipole oriented along the z-axis. Therefore, to simplify the model, the e-beam is represented by one point dipole (labeled as z-dipole) polarized along its trajectory (z-axis) [13,30,43], as shown by the red double-headed arrows in Figure 3. The dipole is placed closest to the Ag NP on its trajectory. There the dipole couples with the LSP most strongly. A black power transmission box is used to record the scattered power of the whole three-body system. Another two yellow transmission boxes are used to record the radiated powers of the z-dipole and q-dipole. A red plane monitor placed over the p-GaN layer is used to collect the power emitted into the air. By default, in the FDTD system, the powers recorded by all monitors in one simulation are automatically normalized to the sum of the powers from all sources ($P_{source}$). For consistency, all calculated powers are re-normalized by multiplying a normalization factor of $P_{source}/P_{0}$, where $P_{0}$ is the radiated power of the q-dipole in a homogeneous environment (here GaN with a refractive index of 2.55).

Strictly, the QW is a two-dimensional structure which spreads all over the x–y plane. Therefore, in the simulation the q-dipole need to be moved on the x–y plane and then the final result can be got from all the simulations with different q-dipole positions through a certain statistical method. If there is no z-dipole in the simulation, the model is able to be simplified by putting the q-dipole right below the Ag NP (Point A’ in Figure 3) due to the symmetry of the whole simulation system. This simplified model has been used in our previous works [29,30,31]. However, once the z-dipole is added into the simulation, the symmetry of the system will be broken unless the z-dipole is placed at Position A. To simplify the model, the q-dipole is placed at such points where the two dipoles interact most strongly. In addition, the coupling between dipoles and LSPs highly depends on the shape of the Ag NP [44,45]. Thereby, we introduce an aspect factor *k* = *h*/*r* to characterize the shape of the Ag NP, where h and r represent the height and radius of the spherical-cap-shaped Ag NP, respectively. Here we set *k* = 1.67 according to the experimental results. The effect of the *k* factor on the coupling of LSPs and dipoles will also be discussed later. In order to find the optimal position to place the q-dipole, the distribution of the electric field intensity near the Ag NP is calculated in a simulation with only the z-dipole and the Ag NP. The z-dipoles are placed at Point A, B, and C individually, and the electrical field profiles at 551 nm on the x–z plane are shown in Figure 4 for each point. The positions with the maximum electric field strength on the QW plane are found at the center of the bottom of the Ag NP (Point A’), the left edge of the bottom of the Ag NP (Point B’) and right below the z-dipole (Point C’) in Figure 4a–c, respectively. Therefore, in the following simulation of the three-body system, the q-dipoles are placed at Points A’, B’, and C’ when the z-dipoles are put at Points A, B, and C, respectively, as shown in Figure 3. This approximation is acceptable since the strength of the z-dipole is much stronger than that of the q-dipole [13].

Figure 5a shows the calculated Purcell factor ($F_{p}$) curves for the q-dipole with the presence of the z-dipole at point B and the q-dipole at point B’. $F_{p}$ also quantifies the increase in SER described by [1,40]:(2)$$F_{p}=\frac{K_{rad}+K_{non}+K_{sp}}{K_{rad}+K_{non}}$$
where $K_{rad}$ and $K_{non}$ are the radiative and non-radiative recombination rates of electron-hole pairs, and $K_{sp}$ is the LSP–QW coupling rate. In FDTD simulations, $F_{p}$ is defined as the radiation power enhancement of a dipole and related to the local density of states (LDOS) [5]. It is observed that there is one peak in the range of 540–560 nm for each $F_{p}$ curve. It is due to the designed shape and size of the Ag NPs. In this case, the SP resonantly couples to the green QWs. The resonant wavelength first blue shifts as φ increases from 0° to 90°, and then it redshifts as φ increases from 90° to 180°. The peak value of $F_{p}$ increases monotonously as φ increases from 0° to 180°. It means the polarized angle of dipole takes an important role in the coupling system of e-beam–LSP–QW [25,27,30]. For comparison, $F_{p}$ for the q-dipole without the Ag NP or the z-dipole is also calculated, as shown in Figure 5b,c. It is found in Figure 5b that all the $F_{p}$ values at 551 nm vary near 1.06 and do not change much as φ changes, which indicates that the direct interaction between the two dipoles is much weaker than the indirect interaction through the LSP of the Ag NP. Therefore, the large values of $F_{p}$ in Figure 5a are attributed to the q-dipole coupling to ebLSP excited by z-dipole rather than coupling to z-dipole directly. Besides this, It’s worth noting that when placing the q-dipole at Point B’ the original symmetry of the surrounding environment for the q-dipole is broken even without the z-dipole compared with Point A’. Because Point A’ corresponds to the center of the round bottom of Ag NP, the electrical field is symmetry as shown in Figure 4a. Figure 5c shows the $F_{p}$ curve of the q-dipole with the Ag NP but without the z-dipole. In this case, there is no symmetry breaking caused by the phase difference between the two dipoles. Accordingly, the $F_{p}$ values for the polarized angle φ and 180° − φ are completely identical. It is seen that $F_{p}$ peak values are larger for *φ* < 90°and smaller for *φ* > 90° without z-dipole than those in Figure 5a. The z-dipole changes the symmetry of the LSP electrical field, which modifies LSP couplings to the different polarized q-dipoles.

All the $F_{p}$ curves at 551 nm varying with different polarized angles in Figure 5a–c are plotted in Figure 5d. When excluding the Ag NP from the system, $F_{p}$ decreases monotonically from 1.141 to 1.005 as φ increases from 0° to 180°. Generally the direct interaction of the orthogonal dipoles is much weak without SP coupling [13]. The small decrease of $F_{p}$ as φ increases is due to the deviation along x-axis between z-dipole and x-dipole in Figure 3. In the case without the z-dipole, the LSP is only excited by the q-dipole. The $F_{p}$ curve at 551 nm shows a trend of increasing first, reaching the maximum of 2.354 at 90° and then decreasing to the start point of 1.999 at 180°. When both the Ag NP and z-dipole are added into the system, the q-dipole couples to the ebLSP and qwLSP simultaneously. The $F_{p}$ curve shows a similar trend of first increasing from 1.616 to 2.457, and then decreasing to 2.383 at 180°. But the maximum appears at 120°. Apparently, in the three-body system, both the z-dipole and Ag NP play an important role in the LSP coupling with the q-dipole. However, the three curves shown in Figure 5d have no simple multiplicative relationship, indicating the interactions of the three bodies are coupled together rather than affected separately. As mentioned above, the ebLSP induced by the z-dipole in this case mainly polarizes along the x-axis and is symmetric about the x-axis. As φ changes, the polarization orientation of the q-dipole changes. Therefore, the interaction between the ebLSP and qwLSP will vary with the change of φ, leading to the variation of LDOS at the q-dipole position [5]. As for Point A, $F_{p}$ changes very little (about 0.01) at 551 nm when the z-dipole is added. Because the LSP symmetry is not broken, $F_{p}$ keeps constant of 3.122 at 551 nm for all the φ. The $F_{p}$ value is larger than the average one of 2.151 at 551 nm in Figure 5a. It implies that the q-dipole can’t have a strong coupling with the LSP at the edge of the bottom of the Ag NP. When the z-dipole is placed at Point C, the $F_{p}$ values at 551 nm vary near 1.1 as φ changes. It is obvious that the dipoles representing e-beam and QW at point C are 90 nm away from the Ag NP, which is out of the evanescent field of the LSP. Both ebLSP and qwLSP are very weak.

Given that the e-beam itself does not radiate light, the energy radiated by z-dipole needs to be subtracted from the three-body (z-dipole-Ag-q-dipole) system. Since the power of the z-dipole and q-dipole is either scattered or dissipated by the Ag NP, the quantum efficiency (QE) for the system can be defined as [38]:(3)$$\eta_{QE}=\frac{\sum_{s}\alpha_{s}\left(\lambda \right)*P_{s}\left(\lambda \right)}{\sum_{s}P_{s}\left(\lambda \right)}$$
where the subscript ‘s’ represents the dipole index in the system, $\alpha_{s}\left(\lambda \right)$ is the scattering rate of the s^th^ dipole and $P_{s}\left(\lambda \right)$ is the radiated power of the s^th^ dipole. Specially, $P_{s}\left(\lambda \right)$ is equavilent to $F_{p}$ for the q-dipole because of the re-normalization. According to the records of the monitors in Figure 3, the summation of the scattered powers, namely $\sum_{s}\alpha_{s}\left(\lambda \right)*P_{s}\left(\lambda \right)$, is recorded by the black monitor box ($P_{bBox}$) while $P_{s}\left(\lambda \right)$ can be obtained using the yellow monitor boxes ($P_{yBox}$). Obviously:(4)$$P_{bBox}=\underset{s}{\sum }\alpha_{s}\left(\lambda \right)*P_{s}\left(\lambda \right)$$

Considering that $\alpha_{s}$ is the characterization of scattering ability of a dipole, it can be written as a function of $P_{s}\left(\lambda \right)$, namely $\alpha_{s}\left(P_{s}\left(\lambda \right)\right)$. Based on the Taylor series, $\alpha_{s}\left(P_{s}\left(\lambda \right)\right)$ can be expanded as:(5)$$\alpha_{s}\left(P_{s}\left(\lambda \right)\right)\approx \alpha_{s}^{\left(0\right)}+\alpha_{s}^{\left(1\right)}\delta P_{s}\left(\lambda \right)$$
where the superscripts of $\alpha_{s}^{\left(0\right)}$ and $\alpha_{s}^{\left(1\right)}$ indicate the zeroth and the first derivative of $\alpha_{s}$. To solve $\alpha_{s}^{\left(0\right)}$ and $\alpha_{s}^{\left(1\right)}$, a perturbation method is used by adding a “small” term into the three-body system. Since $P_{s}\left(\lambda \right)$ is mainly regulated by its amplitude (A), $P_{s}\left(\lambda \right)$ can be exactly written as a function $P_{s}\left(\lambda ,\;A\right)$. In the simulations, a small variation ΔA is added to the q-dipole, that is, changing A to A+ΔA. After running the simulation a second time, $\delta P_{s}\left(\lambda ,\;A\right)$ can be obtained from $P_{s}\left(\lambda ,\;A+\Delta A\right)-P_{s}\left(\lambda ,\;A\right)$ and Equation (4) can be rewritten as:(6)$$P_{bBox}^{\prime }=\underset{s}{\sum }\alpha_{s}^{\prime }\left(\lambda \right)*P_{s}^{\prime }\left(\lambda \right)\approx \underset{s}{\sum }\left(\alpha_{s}^{\left(0\right)}+\alpha_{s}^{\left(1\right)}\delta P_{s}\left(\lambda ,\;A\right)\right)*P_{s}^{\prime }\left(\lambda \right)$$
where the prime (′) indicates the system has been “perturbed”. Then $\alpha_{s}^{\left(0\right)}$ and $\alpha_{s}^{\left(1\right)}$ can be solved by combining the system of linear Equations (4) and (6). To make sure this perturbation method give the correct and convergent results, ΔA should be small enough. Based on our simulation, it is found that the calculation converges well when ΔA/A is less than 10^−4^. Once $\alpha_{s}$ is calculated, the IQE of the q-dipole can be obtained by the equation below [13,31,38]:(7)$$\eta_{IQE}=\frac{F_{p}K_{rad}}{F_{p}K_{rad}+K_{non}}\cdot \frac{P_{scat}}{P_{diss}+P_{scat}}$$
where $P_{scat}$ is the scattered power of the q-dipole, $P_{diss}$ is the power that dissipated by the Ag NP. The ratio $K_{rad}/K_{non}$ is based on the original IQE value of 26%, which is measured in a temperature-dependent PL measurement system. Noticing that the ratio $P_{scat}/\left(P_{diss}+P_{scat}\right)$ represents the scattering rate of the q-dipole, namely the factor α in Equation (4), Equation (7) can be simplified as the following:(8)$$\eta_{IQE}=\frac{F_{p}K_{rad}}{F_{p}K_{rad}+K_{non}}\cdot \alpha$$

Similarly, the power extracted into the air ($P_{rPlane}$, recorded by the red plane monitor) can also be written as an equation like Equation (4) by replacing $P_{bBox}$ with $P_{rPlane}$, and $\alpha_{s}\left(\lambda \right)$ with a new parameter $\beta_{s}\left(\lambda \right)$. Apparently,
(9)$$\beta =\frac{P_{up}}{P_{diss}+P_{scat}}$$
where $P_{up}$ is the power of the q-dipole emitted into the air. Based on the previous derivation, the LEE of the q-dipole can be calculated as below:(10)$$\eta_{LEE}=\frac{P_{up}}{P_{scat}}=\frac{\beta }{\alpha }$$
Afterwards, the EQE is calculated by:(11)$$\eta_{EQE}=\eta_{IQE*}\eta_{LEE}$$

According to the perturbation calculation, the energy scattering and dissipation of the three-body system are obtained individually. Figure 6a–c show the average scattering rate spectra of the q-dipoles for Points A, B, and C, respectively. All the scattering rates are weighted averaged by their $F_{p}$ values with φ from 0° to 180°. For Point A, the average scattering rate at 551 nm is 60.72%, which is larger than that without the z-dipole calculated as 59.81%. For longer wavelength, $F_{p}$ with the z-dipole is much larger than that without the z-dipole, as shown in Figure 6a. At Point A, even though the two dipoles are orthogonal to each other and $F_{p}$ changes little for the cases with and without z-dipole, the dissipation can be suppressed when affected by the ebLSP, which conforms to our previous results [13]. As for Point B, the average scattering rate at 551 nm is 47.36%, indicating more than half of the total radiated power is dissipated by the Ag NP. It is also larger than that without the z-dipole calculated as 44.01%, implying the dissipation is also suppressed when adding the z-dipole. Besides this, it is found that both the curves in Figure 6b have a minimum value close to 550 nm, which means the LSP symmetry breaking leads to the low $F_{p}$ and high energy dissipation. At short wavelength region, there seems a resonant peak with high scattering rates. As for Point C, the scattering rate is more than 95% and changes little from 485 to 635 nm, as shown in Figure 5c. The Ag NP stays too far away from the q-dipole to dissipate the energy. Moreover, $F_{p}$ is about 1.1 and the IQE cannot increase obviously since the LSP coupling to the QW and e-beam is too weak.

Once the scattering rate is obtained, the IQE and LEE of the QW can be calculated by using Equations (8)–(11). Table 1 shows all the calculated results changing the aspect factor k and the positions where the dipoles are placed. At Point A, $F_{p}$ with *k* = 1.67 is much smaller than that with *k* = 1 or 1.33. However, the scattering rate is much larger than those with the smaller *k* value. It indicates that the coupling strength between the q-dipole and LSP becomes weaker, but the dissipation can be suppressed more effectively when k becomes larger. According to Equation (8), the IQE for *k* = 1.67 is much higher than that for *k* = 1 or 1.33. This IQE is the highest one in the entire table as well. Moreover, the LEE for larger k is also higher, especially, more than 3 times for the LEE with *k* = 1.67 than that with *k* = 1. In short, the larger aspect factor provides both higher IQE and LEE. This result conforms to those in our previous work [44]. As for Point B, the aspect factor k has a similar effect on Fp, the scattering rate, IQE and LEE increase significantly as those for Point A. As for Point C, $F_{p}$ is close to 1 and scattering rate close to 100%, which means the weak coupling and low dissipation. When *k* = 1.67, the EQE is 1.18 and 1.54 times for Points A and B compared with that for Point C, which is consistent with the experimental results of 1.26 and 1.52 times. Comparing Points A and B with Point C, the enhancement of EQE at Point B can mainly be attributed to the increase of LEE from 2.945% to 6.506%, and the IQE increase for Point A from 27.49% to 31.76%. It means that both the effects of the resonator and antenna are significant on the light emission enhancement for LSP coupling to the radiators. Reduction of the metal shield and effectively scattering the near-field energy to far-field are the key issues. Although ebLSP is not conventional in practical LEDs, it acts as a useful tool to analyze the metallic nanostructures and orthogonal dipoles interaction. From the perturbation based simulation, the variable separation in the many-body system will clarify the energy transferring processes in the SP coupling to the radiators, which benefits for the future design of high-efficiency LED structures.

## 4. Conclusions

In summary, the LSP–QW coupled sample was fabricated by embedding the Ag NPs into a PhC hole array on green LEDs. Twenty-six percent and fifty-two percent enhancements of the CL intensities are obtained at the center and edge of the Ag NP, respectively compared to the result that e-beam excited QW directly. A perturbation-based simulation method was carried out to analyze the coupling process of the three-body system consisting of a z-dipole, Ag NP and q-dipole. By adding a small perturbation to the three-body system, the individual scattering rate spectra are obtained for the QWs at different points. The calculated result shows that the quantum efficiencies of the QWs (q-dipole) are a function of the azimuthal angle and strongly affected by the ebLSP induced by e-beam. The perturbation model successfully demonstrates the CL results and can be extended to any many-body system consisting of SP and more than one radiators. Using CL technique, the metallic nanostructures and the orthogonal radiators can be optimized to improve the performance of low-efficiency light emitters.

## Figures and Tables

**Figure 1 nanomaterials-10-00913-f001:**
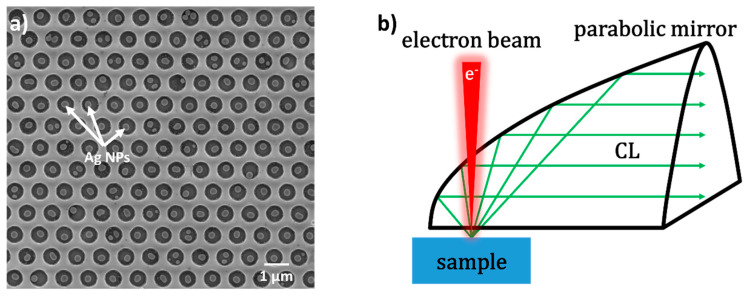
(**a**) Scanning electron microscope (SEM) image of the Ag–photonic crystal (PhC)/quantum well (QW) sample. (**b**) Schematic setup for the cathodoluminescence (CL) measurement.

**Figure 2 nanomaterials-10-00913-f002:**
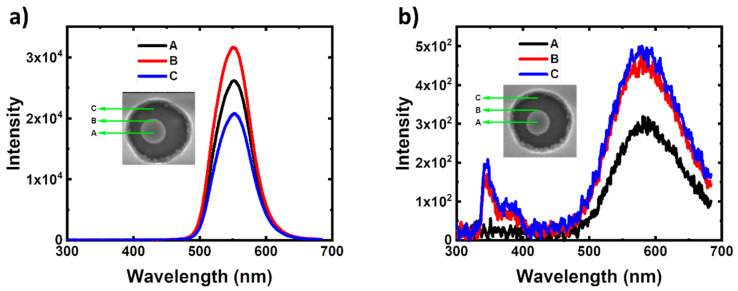
CL spectra measured at Points A, B, and C for (**a**) the Ag–PhC/QW sample and (**b**) Ag–PhC/woQW sample. Their insets show the measurement points in the SEM images.

**Figure 3 nanomaterials-10-00913-f003:**
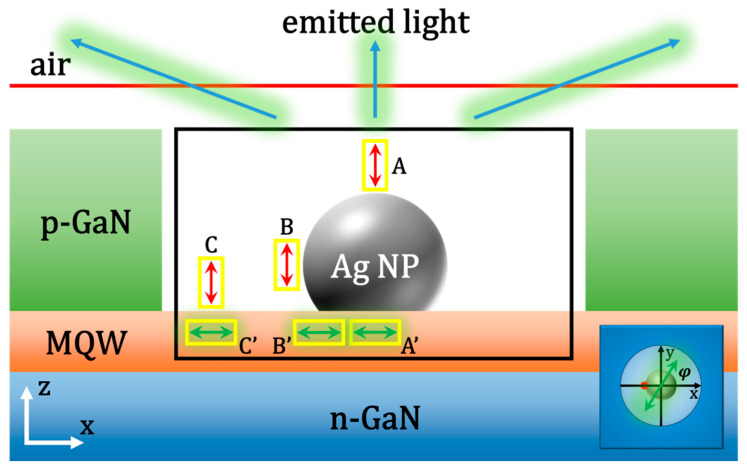
The schematic structure in the three-dimensional (3D) finite difference time domain (FDTD) simulation. The black box monitor is used to collect the scattered power of the whole three-body system. The yellow transmission boxes are used to record the radiated powers of the z-dipole and q-dipole. The red plane monitor is used to record the power emitted from the top surface. The red and green double-headed arrows represent the e-beam and the QW, respectively. The inset shows the polarized angle (φ) of the q-dipole.

**Figure 4 nanomaterials-10-00913-f004:**
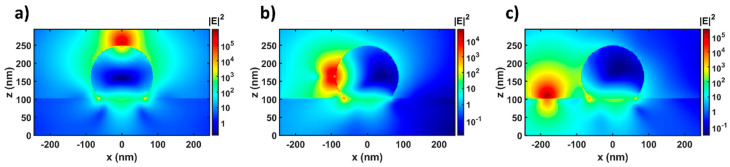
The electrical field profiles at 551 nm on the x–z plane when placing the z-dipoles at Points (**a**) A, (**b**) B, and (**c**) C. All the simulations are run with the aspect factor *k* = 1.67 and without the q-dipoles added.

**Figure 5 nanomaterials-10-00913-f005:**
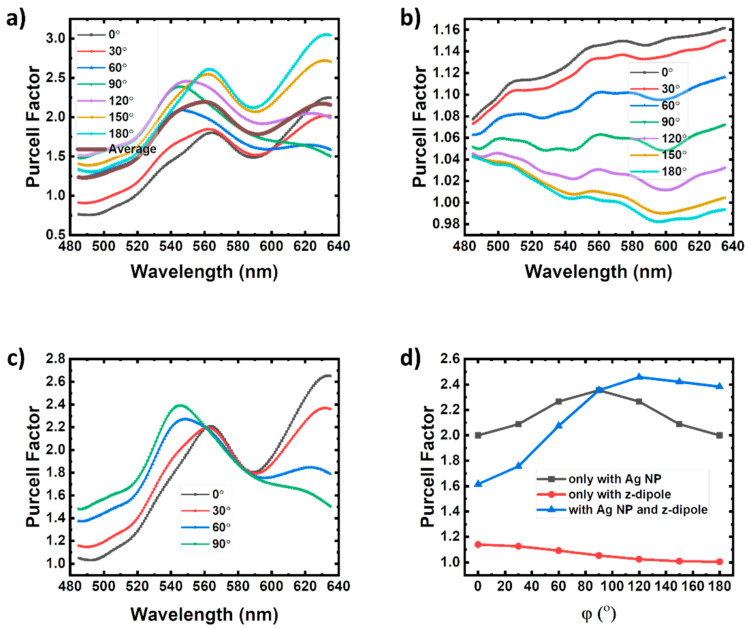
Purcell factors of the q-dipole: (**a**) with both the Ag nanoparticle (NP) and z-dipole, (**b**) only with the z-dipole and (**c**) only with the Ag NP. The z-dipole and q-dipole are placed at Point B and B’, respectively. (**d**) The dependence of the Purcell factors of the q-dipole at 551 nm on different polarized angles φ only with the Ag NP, only with the z-dipole and with both the Ag NP and z-dipole.

**Figure 6 nanomaterials-10-00913-f006:**
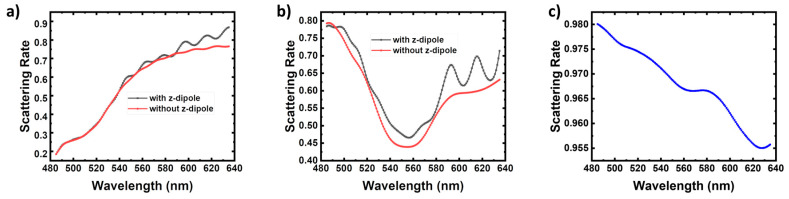
Average scattering rate spectra of the q-dipoles for Points (**a**) A, (**b**) B, and (**c**) C. All the scattering rates are calculated using the perturbation method and weighted averaged by the $F_{p}$ values with φ increasing from 0° to 180°.

**Table 1 nanomaterials-10-00913-t001:** Purcell Factor, Scattering Rate, internal quantum efficiency (IQE), light extraction efficiency (LEE), and external quantum efficiency (EQE) at 551 nm for the q-dipole in the three-body system with the z-dipole placed at different points and different aspect factor k of the Ag NP.

Point	*k*	*F_p_*	Scattering Rate	IQE	LEE	EQE
A	1	7.047	33.64%	23.96%	0.849%	0.203%
	1.33	5.837	43.06%	28.95%	2.658%	0.770%
	1.67	3.122	60.72%	31.76%	3.004%	0.954%
B	1	3.320	34.20%	18.41%	1.192%	0.219%
	1.33	3.105	32.00%	16.70%	4.257%	0.711%
	1.67	2.151	44.39%	19.11%	6.506%	1.243%
C	1	1.114	96.81%	27.23%	2.675%	0.728%
	1.33	1.124	96.99%	27.47%	2.879%	0.791%
	1.67	1.128	96.86%	27.49%	2.945%	0.809%

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
