# Peer review of "Study on Electron-Induced Surface Plasmon Coupling with Quantum Well Using a Perturbation Method"

_nanomaterials, 2020, doi:10.3390/nano10050913_

Round 1

Reviewer 1 Report

The manuscript nanomaterials-798396 reports the analysis of the coupling process between electron beam induced localized surface plasmon (LSP) of Ag nanoparticles (NPs) and quantum well (QW) structures using cathodoluminescence spot scans. The LSP-QW coupling mechanism is investigated by a perturbation calculation making use of a three-dimensional (3D) finite difference time domain (FDTD) method.

This paper is clearly written and presents a detailed interpretation of the numerical calculations obtained from the perturbation-based simulation method of the three-body system comprising high energy electron-beam (e-beam), Ag NP and quantum wells. The work provides sufficient results regarding the perturbation model and the cathodoluminescence experimental data. The comparison between simulated and experimental results is also well described by showing the energy scattering and dissipation of the three-body system. Therefore, this manuscript should be suitable for publication after considering the following recommendations:

In order to emphasize the relevance and new contributions of the investigation, further explanation of the applications of the proposed method should be taking into account by considering the benefits derived from studying the coupling effects of electron-induced surface plasmon with quantum well for improving the performance of low-efficiency light emitters and subsequently enable analytical applications.

Typo errors:

  • Line 24 and line 333: ‘26% and 52% enhancements’, It is recommended to spell out numbers at the start of a sentence.

Reviewer 2 Report

In the paper ‘Study on Electron-Induced Surface Plasmon Coupling with Quantum Well Using a Perturbation Method’ the localized surface plasmon LSP - quantum well QW coupling effect in the three-body system (the e-beam-LSP-QW) is studied. The LSP-QW coupled sample was fabricated by embedding the Ag nanoparticles into a hole array on the GaN-based LED. The enhancements of the cathodoluminescence intensities are obtained in the cases when the e-beam is focused at the center and edge of the Ag nanoparticles.

The obtained results are interesting. The methods adequately described, and he results are clearly presented. I recommend this manuscript be published in Nanomaterials pending minor revisions.

Comments and suggestions:
1) In Fig. 1,a please indicate what is Ag nanoparticles.

2) It might be good if the authors check English (Finite elementary method (FEM),…).

3) The authors should comment on reason why the e-beam can be represented

(a) by one point dipole

(b) polarized along its trajectory (z-axis).
